# Human IgM^hi^CD300a^+^ B Cells Are Circulating Marginal Zone Memory B Cells That Respond to Pneumococcal Polysaccharides and Their Frequency Is Decreased in People Living with HIV

**DOI:** 10.3390/ijms241813754

**Published:** 2023-09-06

**Authors:** Joana Vitallé, Olatz Zenarruzabeitia, Aitana Merino-Pérez, Iñigo Terrén, Ane Orrantia, Arantza Pacho de Lucas, José A. Iribarren, Lucio J. García-Fraile, Luz Balsalobre, Laura Amo, Belén de Andrés, Francisco Borrego

**Affiliations:** 1Immunopathology Group, Biocruces Bizkaia Health Research Institute, 48903 Barakaldo, Spain; olatzena@gmail.com (O.Z.); aitanamerino99@hotmail.com (A.M.-P.); iterren.ofi@gmail.com (I.T.); aneo14@gmail.com (A.O.); laura.amoherrero@osakidetza.eus (L.A.); 2Instituto de Biomedicina de Sevilla (IBiS), Virgen del Rocío University Hospital, CSIC, University of Seville, 41013 Seville, Spain; 3Regulation of the Immune System Group, Biocruces Bizkaia Health Research Institute, 48903 Barakaldo, Spain; mariaaranzazu.pachodelucas@osakidetza.eus; 4Immunology Service, Cruces University Hospital, 48903 Barakaldo, Spain; 5Department of Infectious Diseases, Donostia University Hospital, Biodonostia Health Research Institute, 20014 Donostia-San Sebastián, Spain; joseantonio.iribarrenloyarte@osakidetza.eus; 6CIBER de Enfermedades Infecciosas (CIBERINFEC), Instituto de Salud Carlos III, 28029 Madrid, Spain; lucio.garciafraile@gmail.com; 7Department of Internal Medicine, La Princesa University Hospital, 28006 Madrid, Spain; 8Laboratory of Microbiology, UR Salud, Infanta Sofía University Hospital, 28702 Madrid, Spain; lbalsalobre@ursalud.com; 9Ikerbasque, Basque Foundation for Science, 48009 Bilbao, Spain; 10Immunobiology Department, Carlos III Health Institute, 28220 Madrid, Spain; bdandres@isciii.es

**Keywords:** CD300, CD300a, IgM, IgD, B cell, memory, marginal zone, pneumococcus, polysaccharides, HIV

## Abstract

CD300a is differentially expressed among B cell subsets, although its expression in immunoglobulin (Ig)M^+^ B cells is not well known. We identified a B cell subset expressing CD300a and high levels of IgM (IgM^hi^CD300a^+^). The results showed that IgM^hi^CD300a^+^ B cells were CD10^−^CD27^+^CD25^+^IgD^lo^CD21^hi^CD23^−^CD38^lo^CD1c^hi^, suggesting that they are circulating marginal zone (MZ) IgM memory B cells. Regarding the immunoglobulin repertoire, IgM^hi^CD300a^+^ B cells exhibited a higher mutation rate and usage of the IgH-VDJ genes than the IgM^+^CD300a^−^ counterpart. Moreover, the shorter complementarity-determining region 3 (CDR3) amino acid (AA) length from IgM^hi^CD300a^+^ B cells together with the predicted antigen experience repertoire indicates that this B cell subset has a memory phenotype. IgM memory B cells are important in T cell-independent responses. Accordingly, we demonstrate that this particular subset secretes higher amounts of IgM after stimulation with pneumococcal polysaccharides or a toll-like receptor 9 (TLR9) agonist than IgM^+^CD300a^−^ cells. Finally, the frequency of IgM^hi^CD300a^+^ B cells was lower in people living with HIV-1 (PLWH) and it was inversely correlated with the years with HIV infection. Altogether, these data help to identify a memory B cell subset that contributes to T cell-independent responses to pneumococcal infections and may explain the increase in severe pneumococcal infections and the impaired responses to pneumococcal vaccination in PLWH.

## 1. Introduction

Human CD300a is a transmembrane protein, with an immunoglobulin (Ig)V-like extracellular domain and a cytoplasmic tail containing immunoreceptor tyrosine-based inhibitory motifs (ITIMs), providing the receptor with an inhibitory capacity [1,2]. The relevance of the CD300a molecule in several pathological conditions has been highlighted by multiple studies [3,4,5,6,7]. Among others, CD300a has been related to bacterial, parasitic, and viral infections [4,8,9,10,11,12,13,14,15]. For instance, after cecal ligation and puncture peritonitis in mice, the absence of CD300a induced a more effective bacterial clearance and prolonged mice survival by increasing mast cell inflammatory response and neutrophil recruitment [16,17]. CD300a is broadly expressed on myeloid and lymphoid cells, and its expression is known to be differentially regulated depending on the cell type [11,12,13,14,15,18,19,20,21,22,23]. Regarding B cells, besides some aspects of the expression and function of CD300a on several human B cell subsets, its relevance in B cell leukemogenesis, and its potential relevance in viral infections [23,24,24,25,26], its role is not completely known.

In humans, B cells have a central role in adaptive immune responses, having different contributions depending on the B cell subset [27]. Immune memory is a key factor in the protection from recurring infections and, among their diverse compartments, memory B cell subsets expressing IgM are particularly interesting due to their role against specific pathogens, such as encapsulated bacteria, and also as a source of long-lived memory, as switched memory B cells [28,29,30,31,32,33,34,35,36,37]. In peripheral blood, human circulating IgM memory B cells, mostly characterized by the expression of CD27, can be divided into two subsets according to the expression of IgD: IgM^+^IgD^lo^ and IgM^+^IgD^hi^ B cells [38]. Gene-expression studies revealed that IgM^+^IgD^lo^ are analogous to marginal zone (MZ) B cells and, after different in vitro stimuli, these circulating B cells showed higher proliferation and differentiation to antibody-secreting cells but lower IgG class switching than IgM^+^IgD^hi^ B cells [30,34,37,38,39,40,41,42,43,44,45]. Importantly, the lack of IgM memory B cells has been associated with impaired production of pneumococcal-specific antibodies [46]. Moreover, a reduction in the frequency of this cell subset has been found in people with an increased susceptibility to encapsulated bacterial infections, for example, in congenitally asplenic and splenectomized individuals [46] and patients with common variable immunodeficiencies [47]. A reduction of IgM^+^IgD^lo^ B cells that did not recover with antiretroviral therapy (ART) has been also observed in people living with HIV-1 (PLWH) [28]. Furthermore, PLWH with decreased IgM^+^IgD^lo^ memory B cells displayed impaired pneumococcal IgM vaccination responses [28].

In this work, a previously undetermined human B cell subset was identified, which was positive for CD300a and displayed high levels of IgM. Through a phenotypical characterization and the analysis of the Ig repertoire, we propose that IgM^hi^CD300a^+^ B cells are circulating MZ B cells. Notably, we identified the capacity of the IgM^hi^CD300a^+^ B cell subset to respond to pneumococcal polysaccharides. Lastly, we found a decrease in the frequency of IgM^hi^CD300a^+^ B cells in PLWH compared to healthy donors.

## 2. Results

### 2.1. CD300a Identifies an IgM-Expressing Memory B Cell Subset

CD300a is an inhibitory receptor with the capacity to suppress B-cell receptor (BCR)-mediated B cell activation and proliferation [25]. We and others have previously described that CD300a is differentially expressed among B cell subpopulations so that while naïve cells express low levels of CD300a, memory B cells and plasma cells show variable levels and germinal center (GC) B cells are negative for this receptor [23,25]. However, its expression in IgM^+^ B cells has not been studied so far. When we analyzed the expression of CD300a among human peripheral blood IgM^+^ B cells, we identified a circulating B cell subset expressing CD300a and high levels of IgM (hereafter IgM^hi^CD300a^+^ B cells) (Figure 1A).

In order to characterize the IgM^hi^CD300a^+^ B cell subset, using multiparametric flow cytometry, we performed a comprehensive phenotypical study in peripheral blood samples from healthy donors, comparing them with circulating IgM^+^CD300a^−^ B cells (Figure 1B,C). We observed that all IgM^hi^CD300a^+^ and the majority of IgM^+^CD300a^−^ B cells were CD10 negative and positive for CD20 and CD21 (Figure 1B), indicating that they are mature B cells. However, they differ in the expression levels of the memory marker CD27, whose expression is significantly higher in IgM^hi^CD300a^+^ B cells compared to IgM^+^CD300a^−^ B cells (Figure 1B,C). We also analyzed IgM^hi^CD27^+^ cells and, interestingly, we found that this subset includes both CD300a^+^ and CD300a^−^ cells (Appendix A). We also studied the expression of different receptors on IgM^hi^CD27^+^CD300a^−^ cells compared with IgM^hi^CD27^+^CD300a^+^ cells and the results revealed that, although very similar, they do not express the same levels of CD10, CD20, CD21, CD23, CD25, CD38, IgD, and HLA-DR (Appendix A). This suggests that IgM^hi^CD300a^+^ and IgM^hi^CD27^+^ cells, although overlapping, do not identify the same B cell subset.

The levels of IgD were significantly lower in IgM^hi^CD300a^+^ B cells than in IgM^+^CD300a^−^ cells (Figure 1B,C). On the other hand, we found that the expression levels of CD23, the low affinity receptor for IgE whose expression is reduced upon memory B-cell differentiation [48,49], and CD38, which regulates B cell activation and is highly expressed by antibody-secreting cells [45,50], were significantly lower in IgM^hi^CD300a^+^ B cells than in IgM^+^CD300a^−^ B cells. Conversely, the IgM^hi^CD300a^+^ B cell subset showed higher levels of CD1c, which is highly expressed by the splenic and circulating MZ B cells [45,51,52] (Figure 1B,C). 

We also analyzed the expression of several surface receptors and co-stimulatory molecules related to B cell activation and memory phenotype. We observed that IgM^hi^CD300a^+^ B cells express significantly higher levels of CD25 and CD86, while the expression of HLA-DR was lower than on IgM^+^CD300a^−^ B cells (Figure 1B,C). Finally, we analyzed the expression of CXCR4, CCR6, CXCR3, and CD62L, receptors that play a fundamental role in B cell migration to the lymph nodes, inflammation sites, and different tissues, and whose expression has been described in circulating B cells, including memory and naïve B cells [53,54,55,56,57]. Regarding these homing receptors, we observed a higher percentage of CXCR3-expressing cells in the IgM^hi^CD300a^+^ B cell subset than in the IgM^+^CD300a^−^ subset (Figure 1C). In addition, while IgM^hi^CD300a^+^ B cells expressed slightly higher levels of CCR6, they exhibited lower levels of CXCR4 and similar levels of CD62L than IgM^+^CD300a^−^ B cells (Figure 1C). 

As IgM^+^CD300a^−^ B cells show a high variability in IgM expression, we decided to divide them into IgM^lo^CD300a^−^ and IgM^int^CD300a^−^ B cell subsets (Appendix A) and compared the expression of previously analyzed markers in the three subsets on a new cohort of donors. In addition to the expected continuing increase in IgM expression, we observed a similar trend in CD1c expression, together with an inverse tendency in HLA-DR, CD23, and CXCR5 expression, from IgM^lo^CD300a^−^ to IgM^int^CD300a^−^ to IgM^hi^CD300a^+^ populations (Appendix A), and similar expression of CCR6. Interestingly, IgM^int^CD300a^−^ B cells expressed the highest levels of IgD and CD38, and IgM^hi^CD300a^+^ B cells expressed the lowest levels of IgD (Appendix A). Finally, the IgM^hi^CD300a^+^ B cell subset expressed significantly higher levels of CD27 and CD25 compared to the other two subsets. The expression of CD27 and higher expression of CD86 and CXCR3, together with the lower levels of IgD, indicate that this population displays a memory phenotype. 

In short, our results showed that IgM^hi^CD300a^+^ B cells are CD10^−^CD27^+^IgD^lo^CD21^hi^CD23^−^CD38^lo^CD1c^hi^, suggesting that CD300a could be distinguishing circulating MZ B cells among all IgM-expressing B cells [27,32,43,45]. Furthermore, we showed that this subpopulation is phenotypically distinct from IgM^lo^CD300a^−^ and IgM^int^CD300a^−^ B cell subsets.

### 2.2. Higher Mutation Frequencies of IgH Receptor in IgM^hi^CD300a^+^ B Cells Compared to IgM^+^CD300a^−^ B Cells

As human splenic MZ B cells have been shown to carry somatic mutations [32,58], we characterize the IgH gene repertoire of IgM^hi^CD300a^+^ B cells in order to validate our hypothesis that they are circulating MZ B cells. Sorted IgM^+^CD300a^−^ and IgM^hi^CD300a^+^ B cells were obtained from eight different donors in order to analyze the Ig VH-repertoire. A total of 75.474 sequences from IgM^+^CD300a^−^ and 47.430 sequences from IgM^hi^CD300a^+^ sorted cells were obtained with similar frequencies of functional sequences, although diversity, VH1 rearrangements, and complementarity-determining region (CDR)3 length were decreased in IgM^hi^CD300a^+^ B cells (Table 1 and Figure 2A). In contrast, higher somatic mutation rates were observed in IgM^hi^CD300a^+^ B cells compared to IgM^+^CD300a^−^ B cells (Figure 2B), as described by human splenic MZ B cells [32]. Analysis of the nature of the somatic mutations (replacement, R; silent, S) and their distribution in framework regions (FWRs) and CDRs revealed a preferential distribution in R mutations in CDR regions. Somatic hypermutations produced by the activation-induced cytidine deaminase (AID) preferentially target RGYW and WRCY motifs located at CDRs [59]. As shown in Table 1, both AID-motifs from IgM^hi^CD300a^+^ B cells were more mutated than IgM^+^CD300a^−^ B cells, denoting selection. Antigen selection strength was quantified using the BASELINe algorithm, which compares the expected mutations based on the random distribution of the observed mutations (Figure 2C). As it is shown in the figure with one representative donor, the selection strength was higher in the CDR region than in FWR from IgM^hi^CD300a^+^ B cells.

### 2.3. T Cell-Independent Response in IgM^hi^CD300a^+^ B Cells

It has been proposed that IgM memory B cells generated in the absence of GC give rise to extra-follicular thymus-independent responses and produce natural antibodies [60,61,62,63,64,65,66]. These antibodies, mostly IgM, are found in individuals without known prior antigenic experience [61,67,68], being important in T cell-independent B cell immune responses against encapsulated bacteria [28,47]. To test the functional capacity of IgM^hi^CD300a^+^ B cells, sorted IgM^hi^CD300a^+^ and IgM^+^CD300a^−^ B cell subsets from different donors were stimulated with 10 µg/mL of each pneumococcal polysaccharides (PPS) strains (3, 14, and 17F) for 7 days. We observed that IgM^hi^CD300a^+^ B cells produce higher amounts of IgM in response to PPS than IgM^+^CD300a^−^ B cells, which barely respond to PPS stimulation (Figure 3A, left panel). On the other hand, PPS stimulation did not induce any IgG production (Figure 3A, right panel), suggesting that there was no class switching. We also observed a significantly higher spontaneous production of IgM in IgM^hi^CD300a^+^ cells than in IgM^+^CD300a^−^ B cells in the eleven tested donors (Figure 3B), a fact that has been previously described by other authors regarding MZ B cells [46,69]. Moreover, among circulating B cells, MZ B cells have been shown to be the main producers of IgM after toll-like receptor 9 (TLR9) ligation with CpG [70]. TLR9 expression was higher in IgM^hi^CD300a^+^ than in IgM^+^CD300a^−^ B cells (Figure 3C), which is associated with a higher IgM production in response to CpG class B (ODN 2006) by IgM^hi^CD300a^+^ B cells (Figure 3D, left panel), while there was no IgG production (Figure 3D, right panel).

### 2.4. Lower Frequencies of IgM^hi^CD300a^+^ B Cells in PLWH 

We finally determined if the frequency of this cell subset is altered during disease. Considering that PLWH has an increase in invasive pneumococcal infections as well as an impaired response to the immunization with pneumococcal vaccine compared with healthy people [28,71], we hypothesized that it could be associated with a decrease in the frequency of circulating IgM^hi^CD300a^+^ B cells. In fact, we observed that the percentage of circulating IgM^hi^CD300a^+^ B cells tended to be lower in ART naïve PLWH than in healthy donors, and this difference was statistically significant when PLWH were under ART (Figure 4A). Preliminary results, with a small cohort of donors, showed that there were no significant changes in the phenotype of IgM^hi^CD300a^+^ B cells in PLWH, with the exception of an increased expression of CD38, suggesting a more activated phenotype, and a small decrease in CD27 and CXCR5 expression levels (Appendix A). The percentage of circulating IgM^hi^CD300a^+^ B cells did not correlate with the viral load or patients’ age and years with ART (Appendix A), but there was a significant negative correlation with the time since the diagnosis of HIV infection (Figure 4B). These results, together with those from the functional assays, suggest that IgM^hi^CD300a^+^ B cells could be associated with the impaired responses to pneumococcal vaccination in PLWH and the increase in invasive pneumococcal infections.

## 3. Discussion

In this work, we identified and characterized a previously unknown B cell subset expressing the inhibitory receptor CD300a and high levels of IgM. According to the phenotypic analysis, IgM^hi^CD300a^+^ B cells are CD10^−^CD27^+^CD25^+^IgD^lo^CD21^hi^CD23^−^CD38^lo^CD1c^hi^. On the one hand, the absence of CD10, a marker of immature B cells, indicates that this is a subpopulation of mature B cells. On the other hand, these cells express both CD25 and CD27, molecules whose expression has been previously described in memory cells [34,72], in addition to a higher expression of CD86 and increased frequency of CXCR3^+^ cells than IgM^+^CD300a^−^ cells. IgM memory B cells are approximately 20% of peripheral blood B cells and 50% of all memory B cell compartments. In humans, peripheral blood IgM memory B cells were initially defined as IgM-only and IgM^+^IgD^+^ B cells, both expressing CD27 and representing 5% and 15% of peripheral blood B cells, respectively [32,38,42]. A recent work by Bautista et al. redefined the IgM^+^IgD^+^ population into two subsets, IgM^hi^IgD^lo^ and IgM^lo^IgD^hi^ B cells, which exhibit different phenotypes and functions in vitro. IgM^hi^IgD^lo^ subset express higher levels of CD27, CD69, CD1c, and CD45RB and lower levels of CD38, IL21R, B-cell activating factor receptor (BAFFR), CD23, CD184, and CD5 compared to IgM^lo^IgD^hi^ [38]. Similarly, Sanz et al. classified the IgM^hi^IgD^lo^ subset as memory unswitched IgM^hi^IgD^lo^CD1c^+^ compared to the IgM^+^IgD^−^, defined as pre-switched [45]. Furthermore, they described that this unswitched IgM^hi^IgD^lo^ subset is the MZ equivalent and has natural memory properties [45]. In our analysis, IgM^hi^CD300a^+^ B cells express lower levels of IgD, CD23, and CD38, together with higher levels of CD27 and CD1c, and represent around 5% of peripheral blood B cells, suggesting that this subset could correspond to the IgM^hi^IgD^lo^ [38] or unswitched [45] memory B cells. In fact, the absence of the CD23 marker, a molecule that drives proliferation and survival during early stages of GC reactions, together with a high CD1c and low IgD expression, characteristic of cells located in the splenic MZ, suggests that these cells could be circulating memory IgM^+^ B cells from the splenic MZ [32,40,43,45,52,73,74,75,76]. In line with this notion is the observation that the numbers of this memory IgM^+^ B cell subset in blood are reduced after splenectomy [77] and in children younger than 2 years of age, when mature MZ B cells are also absent in the spleen [78,79,80]. 

To determine the potential of this B cell subset to migrate and interact with other cell subsets, such as T cells, we analyzed the expression levels of co-stimulatory molecules and chemokine receptors that regulate leukocyte trafficking by promoting the homing to inflammation sites (CXCR3), to secondary lymphoid tissues (CD62L) or to bone marrow (CXCR4) [53,56,81]. In comparison with IgM^+^CD300a^−^ cells, the IgM^hi^CD300a^+^ B cell subset expresses lower levels of HLA-DR [82] and CXCR4, together with higher levels of CD86, a CD4^+^ T cell co-stimulatory molecule constitutively expressed on memory B cells [83], and CXCR3, a receptor expressed on GC B cells which facilitates the migration to the dark zone [84]. This differential expression pattern suggests that the IgM^hi^CD300a^+^ B cell subset may have an increased potential to migrate to inflammation sites and a role in the co-stimulation of T cells. Nevertheless, more studies are required to fully characterize this B cell subset regarding these latter properties.

Human splenic MZ is a unique B-cell compartment that contains B cells with a high surface expression of IgM and complement receptor 2 (Cr2 or CD21) and exhibits a fast activation and Ig secretion in response to blood antigens [85]. We found that IgM^hi^CD300a^+^ B cells harbor a higher mutation rate and usage of the IgH-VDJ genes than the IgM^+^CD300a^−^ counterpart. Furthermore, the shorter CDR3 AA length from IgM^hi^CD300a^+^ B cells compared to IgM^+^CD300a^−^ together with the predicted antigen experience repertoire, aligns this novel B cell compartment with a memory phenotype [86]. The origin of IgM memory B cells is under discussion, and some studies have proposed that IgM^+^IgD^+^ circulating MZ or unswitched IgM memory B cells are generated from GC reactions as is substantiated by high mutation rate of IgHV genes and high expression of Bcl6 gene [42]. Other findings suggest that MZ B cells also carry somatic hypermutation and that these cells are generated through a GC-independent mechanism [29,31,42,87], probably involving TLR-mediated signals [31,70]. In this regard, it has been demonstrated that IgM memory B cells can be generated from transitional B cells through in vitro stimulation with the TLR9 ligand CpG class B (ODN 2006), which induces proliferation, increased AID expression, and acquisition of somatic mutations. This process led to the differentiation into IgM memory cells and the production of IgM natural antibodies directed against capsular polysaccharides of bacteria such as *Streptococcus pneumoniae* [70,87,88]. On this point, our data show an increased presence of the AID motifs (RGYW and WRCY) in IgM^hi^CD300a^+^ B cells. This capacity to accomplish T cell-independent responses against encapsulated bacteria or natural memory has been described by several authors [30,31,42,45,89]. Our in vitro experiments with sorted IgM^hi^CD300a^+^ B cells revealed that this population produces IgM in response to pneumococcal polysaccharides and CpG class B (ODN 2006), and express higher levels of TLR9, corroborating their ability to perform T cell-independent responses.

It has been proposed that, in contrast to switched memory B cells that embody the highly specific adaptive memory, IgM memory B cells may instead act as a first line of defense [29]. Capsular polysaccharides are T cell-independent B cell antigens, and due to their capacity to induce protective antibody responses, are used for vaccination [30,90]. In fact, an altered response to polysaccharide vaccines, as well as an increased susceptibility to bacterial infections has been described in different groups of people with a reduced frequency of IgM memory B cells, e.g., newborns [30,31,42,46,47,91,92], congenitally asplenic and splenectomized individuals [29,46] or patients with common variable immunodeficiency [47,93]. In this regard, our results showing that the IgM^hi^CD300a^+^ B cells produce IgM in response to pneumococcal polysaccharides suggest that this subset may have an important role against bacterial infections. The exact mechanism of how PPS activates IgM^hi^CD300a^+^ B cells to induce the secretion of IgM is not known. It has been shown that PPS complexed with C3d is able to bind to CD21 [94]. Therefore, it may be possible that PPS/C3d simultaneously engages the CD19/CD21 complex and the BCR, regulating the signaling threshold [95]. On the other hand, a TLR-dependent mechanism may also induce the secretion of IgM. In fact, several TLRs such as TLR2, 4, and 9 are known to contribute to the immune recognition of S. pneumoniae [96] and IgM^hi^CD300a^+^ B cells expressed high levels of TLR9. Certainly, in-depth studies are necessary to understand the mechanism underlying PPS-induced IgM secretion by IgM^hi^CD300a^+^ B cells. On the other hand, it remains to be investigated whether the inhibitory receptor CD300a could play a role in the regulation of IgM production by these cells, as it was previously observed regarding BCR-mediated B cell activation [25].

Another circumstance characterized by impaired pneumococcal IgM vaccination responses and increased susceptibility to bacterial infections is HIV-1 infection. PLWH has previously been reported to have decreased levels of IgM memory B cells [28,71], as well as altered CD300a receptor expression in certain B cell subpopulations [25]. When we analyzed the percentage of IgM^hi^CD300a^+^ B cells in PLWH, we observed a decrease compared to values observed in healthy individuals, especially in those PLWH under ART treatment. These results are in line with previous studies, where the percentages of circulating memory IgM B cells observed in PLWH were significantly lower than in healthy donors, levels that were not reversed by ART [28,97]. Nevertheless, it would be interesting to further study whether the differences observed between PLWH under ART and non-treated ones are due to the treatment or to the time elapsed since the infection. In our cohort, the time since the diagnosis of HIV infection is significantly higher in PLWH under ART than in non-treated individuals, and negatively correlates with the percentage of IgM^hi^CD300a^+^ B cells. In contrast, no correlation was observed with years of ART, suggesting that the decrease in this B cell subset is more related to the time of infection than the treatment. Otherwise, considering the ability of IgM^hi^CD300a^+^ B cells to produce IgM antibodies, we believe that it would be interesting to analyze the frequency of this B cell subset in patients with selective IgM deficiency, which present different infections (bacteria, viruses, fungi, and protozoa) [69,98].

There are no clear markers to delineate IgM memory B cells. Here we propose CD300a as a marker for the identification, among IgM^+^ B cells, of circulating MZ memory B cells with the capacity to respond to encapsulated bacteria such as *Streptococcus pneumoniae.* Altogether, our results might contribute to a better characterization of IgM memory B cells and their function against encapsulated bacteria, demonstrating that CD300a could be a good marker for the identification of MZ B cells among the circulating IgM memory B cells. Importantly, CD300a might be a useful marker to easily analyze the frequency of IgM memory B cells in PLWH, which may allow us to predict a higher susceptibility to pneumonia or a lower vaccine response against pneumococcus in these patients. Future studies, including single-cell gene profiling and more thorough analysis of T cell-independent responses in humans, may help to decipher the complexity of the unswitched memory B cell population and its contribution to immune defenses.

## 4. Materials and Methods

### 4.1. Subjects and Samples

In this study, freshly isolated peripheral blood mononuclear cells (PBMCs) from adult healthy donors (n = 28) and cryopreserved PBMCs from healthy donors (n = 20) were analyzed. Blood samples (buffy coats) from healthy donors were collected through the Basque Biobank. The Basque Biobank complies with the quality management, traceability, and biosecurity, set out in the Spanish Law 14/2007 of Biomedical Research and in the Royal Decree 1716/2011. This study was approved by the Basque Ethics Committee for Clinical Research (PI2016112; version 3). All donors provided written and signed informed consent in accordance with the Declaration of Helsinki. PBMCs were freshly isolated from healthy donors’ blood in our laboratory by Ficoll Paque Plus (GE Healthcare, Chicago, IL, USA) density gradient centrifugation.

In addition, cryopreserved PBMCs from healthy donors (n = 20), ART naïve PLWH (n = 20), and PLWH under ART (n = 22) were provided by the HIV Biobank that works together with the CoRIS cohort (Appendix A). Samples were processed following current procedures and frozen immediately after their reception. All patients participating in the study gave their informed consent and protocols were approved by institutional ethical committees.

All HIV-1 infected patients were asymptomatic, were not co-infected with hepatitis C virus, and had more than 200 CD4+ T cell/mm^3^ when the sample was obtained, and the patients had never been diagnosed with AIDS. ART naïve HIV-1 infected subjects had detectable viremia (>10,000 HIV-RNA copies/mL) and they had never been treated with ART, while patients under ART had undetectable viremia and had been treated with ART at least for 6 months. Clinical data of HIV-1 infected patients, which are shown in Table 2, were obtained from the CoRIS database.

### 4.2. Flow Cytometry

B cell subsets were phenotypically characterized by flow cytometry. For that, the following fluorochrome-conjugated monoclonal mouse anti-human antibodies were used: PerCP-Cy5.5 anti-CD10 (HI10a), PE-Cy7 anti-CD20 (2H7), APC anti-CD38 (HIT2), BV421 anti-CD21 (B-ly4), BV510 anti-CD19 (SJ25C1), PerCP-Cy5.5 anti-HLA-DR (G46-6), APC anti-CD25 (M-A251), BV421 anti-CD86 (2331 (FUN-1)), PerCP-Cy5.5 anti-CXCR3 (IC6), PE-Cy7 anti-CXCR4 (12G8) and BV421 anti-IgD (IA6-2) from BD Biosciences (Franklin Lakes, NJ, USA); APC-Cy7 anti-CCR6 (GO34E3), BV421 anti-CD62L (DREG-56), PE-Cy7 anti-CD1c (L161), PerCP-Cy5.5 anti-CD23 (EBVCS-5) and BV421 anti-TLR9 (S16013D) from Biolegend; FITC anti-IgM (SA-DA4) and APC-ef780 anti-CD27 (O323) from eBioscience (San Diego, CA, USA); and PE anti-CD300a (E59.126) from Beckman Coulter. For extracellular staining, 10^6^ PBMCs were incubated with the specific antibodies for 30 min at 4 °C in the dark. Cells were washed with PBS containing 2.5% of bovine serum albumin (BSA) (Sigma-Aldrich, St. Louis, MO, USA), were resuspended in 350 µL of PBS and then samples were acquired within a FACSCanto II flow cytometer (BD Biosciences). For the detection of TLR9, after extracellular staining, PBMCs were washed with PBS containing 2.5% of BSA and were permeabilized and fixed using Cytofix/Cytoperm Plus Kit (BD Biosciences) following the manufacturer’s instructions. Then, they were incubated with BV421 anti-TLR9 for 30 min at 4 °C in the dark. 2 × 10^6^ cryopreserved PBMCs from both healthy donors and PLWH were also incubated with LIVE/DEAD Fixable Near-IR Dead Cell Stain reagent (Invitrogen, Waltham, MA, USA) before the extracellular staining to detect dead cells following the manufacturer´s protocol. Lastly, cells were washed, resuspended in 250 µL of PBS, and acquired in an LSRFortessa X-20 flow cytometer (BD Biosciences). For analysis, lymphocytes were electronically gated according to their forward (FSC-A) and side scatter (SSC-A) parameters. Next, singlets were selected by SSC-A versus SSC-H. Then, B cells were identified by the expression of CD19, and subsets were selected based on the expression of IgM and CD300a receptors (IgM^hi^CD300a^+^) or (IgM^+^CD300a^−^) (See Figure 1A and Appendix A). 

For flow cytometry data analysis FCS 3.0 files were exported from the FACSDiva software v8.0.3 and imported into FlowJo v.10.7.1. for subsequent analysis. The following plug-ins in FlowJo were used: DownSample (1.1) and Uniform Manifold Approximation and Projection (UMAP). Manual and automated analyses were performed. For the automated analysis, events were first downsampled to 7000 cells from the gate of interest (CD19+ B cells) across all samples. Then, downsampled populations were concatenated for the analysis. UMAP was run using the marker expression indicated in each figure and represented as a heatmap and density plot.

### 4.3. Cell Sorting and RNA Extraction 

B cells were enriched from freshly isolated PBMCs (300–400 × 10^6^ cells) by negative selection using the B cell Isolation Kit II Human (Miltenyi Biotec) following the manufacturer’s protocol. 20–30 × 10^6^ B cells were stained for 15 min at room temperature in the dark with the anti-IgM and anti-CD300a antibodies mentioned above and PE-Cy7 anti-CD19 (SJ25C1) from BD Biosciences. Cells were washed with PBS containing 2.5% of BSA and were resuspended at a concentration of 5 × 10^6^ cells/mL for cell sorting. IgM^hi^CD300a^+^ and IgM^+^CD300a^−^ B cells were sorted with a FACSJazz flow cytometer (BD Biosciences) in the Achucarro Basque Center for Neuroscience. Post-sorted analysis yielded an efficiency of over 95%. Some sorted cells were used for functional assays and other cells for RNA extraction and analysis of VH repertoire. The RNA was extracted from sorted cells using RNeasy Plus Micro Kit from Qiagen (Hilden, Germany) following the manufacturer’s instructions and RNA samples were stored at −80 °C. 

### 4.4. Analysis of the VH Repertoire

The VH repertoire of the IgM^hi^CD300a^+^ and IgM^+^CD300a^−^ B cell subsets was analyzed using state-of-the-art generation sequence methodology in the National Center of Microbiology, Carlos III Health Institute (Majadahonda, Madrid, Spain) as described [99]. Briefly, PCR amplifications to analyze VDJ-C were performed using 1U Supreme NZYTaq II DNA polymerase from NZYTech (Lisboa, Portugal) in a PTC-200 DNA Engine cycler from Bio-Rad (Hercules, CA, USA). Rearranged alleles were amplified using a degenerate V primer (comprising IGHV1 to IGHV7 families) including multiple identifier sequences (MID) and CH-specific primers in separate tubes. Amplicons were prepared with the Nextera XT Index kit v2 kit from Illumina Inc. (San Diego, CA, USA) as described by the manufacturer. Libraries were obtained by 500 bp paired-end sequencing on the MiSeq platform (Illumina). Bioinformatic processing included preprocessing with VDJPipe [100] (sequence fusion by filtering with a minimum average quality of 35, a maximum homopolymer of 20, and collapsing into files with total fasta sequences) using the VDJ-server Release 1.1.2 (https://vdjserver.org/). Once the sequences had been successfully matched, they were submitted to IMGT/HighV-Quest Release 3.4.17 [101] for annotation of the CDR3 and full-length VDJ regions. The IMGT output files were analyzed in ARGalaxy [102].

### 4.5. Functional Assays

Pneumococcus- and CpG-induced antibody secretion was determined. B cell enrichment and the sorting of IgM^hi^CD300a^+^ and IgM^+^CD300a^−^ B cell subsets were carried out as explained above. Then, 2 × 10^5^ cells/well were cultured in 200 µL of complete RPMI (cRPMI) in 96-well plates. cRPMI is composed of RPMI 1640 medium supplemented with GlutaMAX (Gibco), 10% Fetal Bovine Serum (FBS) HyClone from Fisher Scientific (Madrid, Spain), 1% non-essential amino acids (Gibco), 1% Sodium Pyruvate (Gibco) and 1% Penicillin-Streptomycin (Gibco). Purified B cells were stimulated with 3 µM of CpG class B (ODN 2006) (InvivoGen, San Diego, CA, USA) or 10 µg/mL of each pneumococcal polysaccharide (strains 3, 14, 17F) (ATCC) for 7 days at 37 °C. A non-stimulated condition was also included as a negative control. Then, the supernatant was collected and stored at −20 °C. The total IgM secretion was measured using an IgM Human ELISA Kit (Invitrogen) following the manufacturer’s protocols. The pneumococcus-specific IgG was analyzed in the Immunology Service of Cruces University Hospital using the VaccZyme^TM^ Anti-PCP IgG Enzyme Immunoassay Kit (The Binding Site Group, Birmingham, UK) following the manufacturer’s protocol. The positive control was a specific serum provided by the manufacturer.

### 4.6. Statistical Analysis and Data Representation

GraphPad Prism v.8.4 was used for graphical representation and statistical analysis. Normality tests were performed to determine the distribution. The Wilcoxon matched-pairs test was applied to study the differences between IgM^+^CD300a^−^ and IgM^hi^CD300a^+^ B cell subsets. The comparison of 4 parametric groups of data was performed using a one-way ANOVA test and 2 parametric groups of data were compared using a paired student *t*-test. For the analysis of VH-IgM repertoires, statistical significance was determined using a *t*-test or Mann–Whitney non-parametric test. To compare healthy donors, ART-naïve HIV-1 infected subjects, and patients under ART, the Kruskal–Wallis test was used. Finally, the Pearson or Spearman tests were performed in correlation studies with parametric or non-parametric data, respectively.

## Figures and Tables

**Figure 1 ijms-24-13754-f001:**
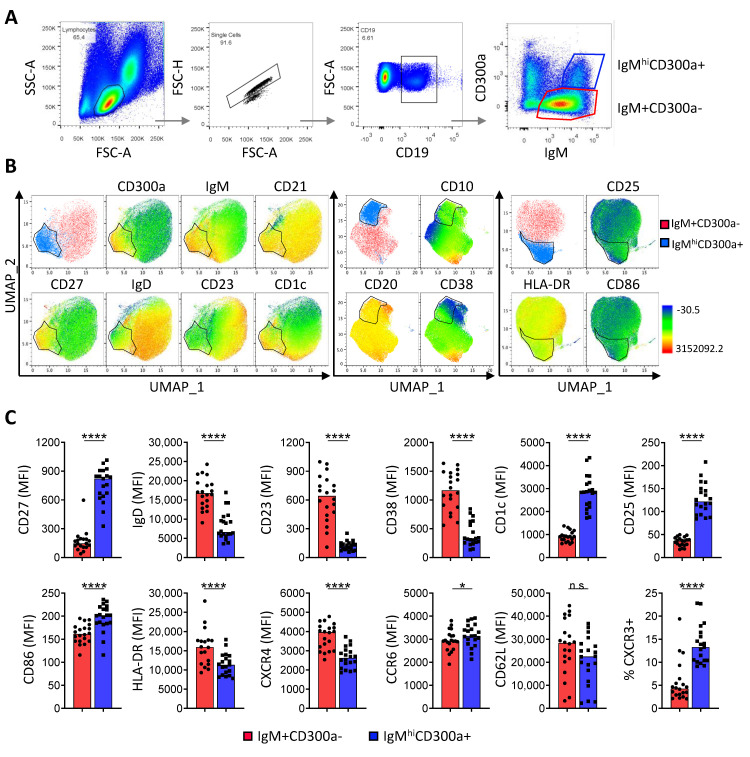
IgM^hi^CD300a^+^ B cell phenotype from peripheral blood of healthy donors. (**A**) Pseudocolor plots showing the gating strategy to distinguish IgM^+^CD300a^−^ (red) and IgM^hi^CD300a^+^ (blue) B cell subsets from peripheral CD19^+^ B cells from healthy donors. (**B**) Three UMAP projections defining IgM^+^CD300a^−^ (red) and IgM^hi^CD300a^+^ (blue) B cell subsets and showing the relative expression of CD300a, IgM, CD10, CD20, CD21, CD27, IgD, CD23, CD38, CD1c, CD10, CD20, CD38, CD25, CD86, and HLA-DR. (**C**) Dot-bar graphs showing the median fluorescence intensity (MFI) of CD27, IgD, CD23, CD38, CD1c, CD25, CD86, HLA-DR, CXCR4, CCR6, and CD62L and the percentage of CXCR3^+^ cells in IgM^+^CD300a^−^ (red) and in IgM^hi^CD300a^+^ (blue) B cell subsets. Data are presented as the mean of 20 individual donors. Statistical significance was determined using a *t*-test or Wilcoxon matched-pairs signed rank test. * *p* < 0.05, **** *p* < 0.0001, ns: not significant.

**Figure 2 ijms-24-13754-f002:**
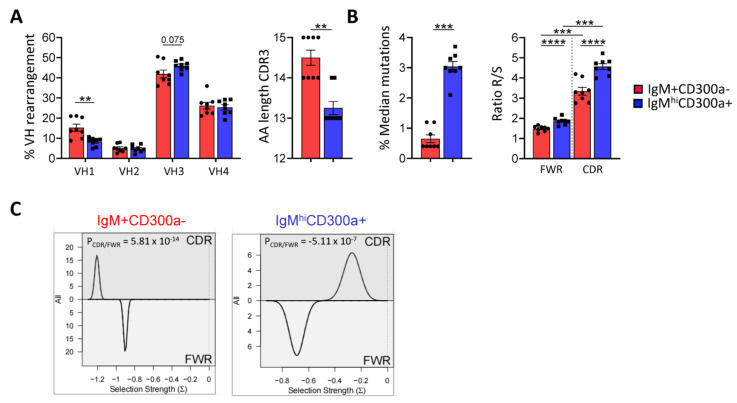
Higher mutation frequencies of IgH receptors in IgM^hi^CD300a^+^ B cells compared to IgM^+^CD300a^−^ B cells. VH-IgM repertoires from sorted cells obtained from eight donors were determined after specific RT-PCR specific amplification. The sequences were processed, cleaned, paired, and analyzed as described in Section 4 and a summary was depicted in Table 1. Each dot represents the repertoire of a single sorted population, IgM^+^CD300a^−^ (red) and IgM^hi^CD300a^+^ (blue). (**A**) Left, frequencies of VH-1,-2,-3 and -4 rearrangements on both cell sorted subsets. Right, AA length of CDR3. (**B**) Left, median frequencies of total mutations. Right, ratios obtained of replacement mutations versus silent mutations (R/S) on FWRs and CDRs regions. (**C**) Selection strength representations using the BASELINe algorithm from a representative donor (#1) on both cell-sorted populations, IgM^+^CD300a^−^ (left) and IgM^hi^CD300a^+^ (right). Data are presented as the mean of 8 healthy donors for each cell compartment. Statistical significance was determined using a *t*-test or Mann–Whitney non-parametric test. ** *p* < 0.01, *** *p* < 0.001, **** *p* < 0.0001.

**Figure 3 ijms-24-13754-f003:**
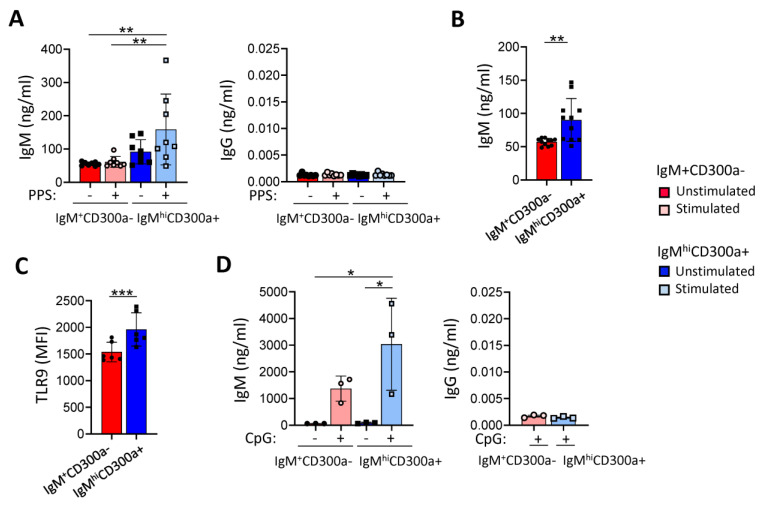
Higher production of IgM in response to PPS and CpG class B (ODN 2006) in IgM^hi^CD300a^+^ B cells than in IgM^+^CD300a^−^ B cells. (**A**) Dot-bar graphs showing the concentration (ng/mL) of IgM (left panel) and IgG (right panel) produced by sorted IgM^+^CD300a^−^ (red) and IgM^hi^CD300a^+^ (blue) B cells after 7 days in the absence (−) or presence (+) of pneumococcal polysaccharides (PPS) (strains 3, 14, 17F). Each dot represents a donor (n = 8). (**B**) Dot-bar graph showing the concentration (ng/mL) of IgM produced by unstimulated sorted IgM^+^CD300a^−^ (red) and IgM^hi^CD300a^+^ (blue) B cells. Each dot represents a donor (n = 11). (**C**) Dot-bar graph showing the MFI of TLR9 in IgM^+^CD300a^−^ (red) and IgM^hi^CD300a^+^ (blue) B cells. Each dot represents a donor (n = 6). (**D**) Dot-bar graphs showing the concentration (ng/mL) of IgM (left panel) and IgG (right panel) produced by sorted IgM^+^CD300a^−^ (red) and IgM^hi^CD300a^+^ (blue) B cells after 7 days in the absence (−) and presence (+) of CpG class B (ODN 2006) (3μM). Each dot represents a donor (n = 3). Data are presented as the mean ± SD. Statistical significance for multiple group comparisons was determined using one-way ANOVA and for single-group comparisons was determined using paired *t*-test. * *p* < 0.05, ** *p* < 0.01, *** *p* < 0.001.

**Figure 4 ijms-24-13754-f004:**
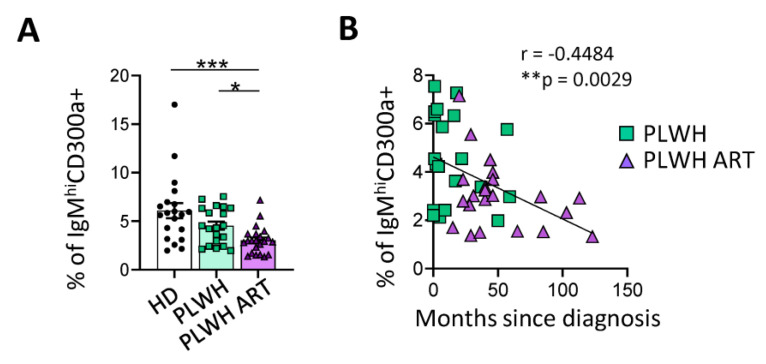
Decreased frequency of IgM^hi^CD300a^+^ B cells in PLWH. (**A**) Dot-bar graphs showing the percentage of IgM^hi^CD300a^+^ B cells in healthy donors (HD) (white), ART-naïve PLWH (HIV) (green), and PLWH under ART (PLWH ART) (purple). Each dot represents a donor (n = 20, 20, 22), mean ± SEMs are shown. Statistical significance was determined by the Kruskal–Wallis test. * *p* < 0.05, *** *p* < 0.001, ns: not significant. (**B**) Graphs showing the association between the percentage of IgM^hi^CD300a^+^ with the months since HIV infection in ART naïve PLWH (PLWH, green squares) and PLWH under ART (PLWH ART, purple triangles). A Pearson test was performed, r = −0.448, ** *p* = 0.0029.

**Table 1 ijms-24-13754-t001:** Summary of the analyzed sequences.

	IgM^+^CD300a^−^						IgM^hi^CD300a^+^					
Donor	Sorted Population	Num. Seq.	Seq Functionality % (Total/Functional)	Seq Diversity % (Unique/Functional)	RGYW %	WRCY %	Sorted Population	Num. Seq.	Seq Functionality % (Total/Functional)	Seq Diversity % (Unique/Functional)	RGYW %	WRCY %
#1	126	18,670	93.15	91.92	15.1	10.8	127	967	83.87	62.56	20.8	14.5
#2	128	12,121	94.1	93.51	14.9	12.6	129	114	98.25	71.93	18.6	18.1
#3	130	9341	93.07	89.63	14.8	10.8	131	8446	93.84	89.78	20.9	16.4
#4	132	10,137	94.54	91.73	19.3	14.6	133	4396	93.63	91.22	21.2	15.9
#5	134	9047	94.86	93.03	19.2	14.8	135	11,449	93.85	90.81	21.7	16.1
#6	136	7226	94.77	92.29	14.9	12	137	9514	93.26	89.1	21.4	15.8
#7	138	4196	91.2	90.36	16.1	11.8	139	9041	92.01	89.45	21.3	16
#8	140	4736	92.8	89.25	12.3	10.9	141	3503	90.24	82.47	22.7	15.5
	**TOTAL**	**75,474**					**TOTAL**	**47,430**				
	**Average**		93.5 ± 0.4	91.47 ± 0.4	15.83 ± 0.8	12.29 ± 0.5	**Average**		92.37 ± 1.4	83.42 ± 3.7 *	21.08 ± 0.4 ****	16.05 ± 0.3 ****

Samples obtained from PBMCs of eight healthy donors were sorted on the basis of CD19+IgM^hi^CD300a− or CD19+IgM^hi^CD300a+. After extraction of RNA, conversion to cDNA and VHμ-PCR were performed as described in Section 4. A summary of the sequences obtained by NGS is presented in the table. Statistical significance for the single-group comparison was determined using a paired *t*-test. * *p* < 0.05, **** *p* < 0.0001.

**Table 2 ijms-24-13754-t002:** Clinical data of PLWH.

	Naïve PLWH		PLWH on ART		
	Median	Range (Min–Max)	Median	Range (Min–Max)	*p*-Value
Sex	Male: n = 20 Female: n = 0	-	Male: n = 20 Female: n = 2	-	
Age (years)	27.5	(22–49)	38.5	(25–60)	*
Months since HIV+	6	(0–59)	40	(15–123)	****
ART (years)	-	-	1	(1–5)	****
Viral load (RNA copies/mL)	18,245	(10,471–125,892)	<20	-	****
CD4+ T cells/mm^3^	502.5	(292–995)	570	(344–1157)	
B2M (mg/L)	2.16	(1.28–3.05)	2.15	(1.47–3.58)	
CRP (mg/L)	1.24	(0.15–7.22)	1.56	(0.17–8.43)	

Mann–Whitney test. * *p* < 0.05, **** *p* < 0.0001.

## Data Availability

Not applicable.

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
