# Peer review of "Human IgMhiCD300a+ B Cells Are Circulating Marginal Zone Memory B Cells That Respond to Pneumococcal Polysaccharides and Their Frequency Is Decreased in People Living with HIV"

_ijms, 2023, doi:10.3390/ijms241813754_

Round 1

Reviewer 1 Report

In this study, Vitalle and colleagues characterized CD300a expressing IgMhi B cells in healthy individuals as well as in people living with HIV. They identified B cell subsets based on CD300a and IgM expression levels and performed phenotypic, VH repertoire and functional analysis. 

They demonstrated that IgMhiCD300a+ B cells have the marginal zone memory B cell features with higher Ig mutation rate and VDJ gene usage as well as shorter CDR3 AA length compared to IgM+CD300a- B cells. The authors showed that these cells secret higher amounts of IgM when stimulated with pneumococcal polysaccharides or TLR9 agonist compared to IgM+CD300a- cells. They also provide evidence that the frequency of IgMhiCD300a+ B cells is significantly lower in ART-treated people living with HIV compared to healthy individuals and it is inversely correlated with duration of HIV infection.

This study is relevant as identifying additional markers to better characterize IgM memory B cells will improve our ability to define their roles in vaccine responses or in pathological conditions. 

Comments:

1.     The data as presented cannot discriminate between CD300+ and CD300- cells among CD27+IgMhi B cells. Do all CD27+IgMhi B cells express CD300a or is CD300a-negative counterpart also presented? It would be very helpful if all data analysis included CD27+IgM+CD300B cells if they were identified. This would add needed depth to the study.

2.     HIV infection is associated with numerous phenotypic abnormalities of B cells, including the over-representation of activated, exhausted B cells associated with HIV-viremia. Considering the heterogeneity of memory B cells from ART-naive individuals living with HIV, it would be beneficial if authors included more detailed analysis of IgM+CD300+ B cells among all memory B cell subsets.  

3.     Figure 1B shows the UMAP plots with marker expression levels. The scales need to be adjusted as it is very difficult to distinguish between positive and negative populations for most of the markers. 

Reviewer 2 Report

This is a well written paper on molecular and cellular Immunity, however I must decline the review based on my not so strong background in Immunology. I don't fell totally qualified to review the paper, and I highly recommend a reviewer with a stron background in cellular immunology.

Author Response

We thank the reviewer for his/her comment.

Reviewer 3 Report

I congratulate the authors, the article is very interesting, the study is very well designed and very well done. The results, too, and discussion are solid and very well described. The research and conclusions are overall excellent. I have, therefore, only a few minor observations.

1.       In the results section the authors describe the phenotype of the IgMhi/CD300a+ subset. In the work, the authors isolated this subset and stimulated it for 7 days with PPS, observing an increase in IgM production. Did the authors done on the stimulated cells a post-stimulus phenotyping? and if yes, a change in the expression of some marker associated with the activation/production of antibodies has been observed ?? for example the transition from CD21 hi CD38 low cells to CD21 low CD38 low (activated) cells ?? if yes it would be interesting And the results should be added to make the work even stronger.

2.       In the results section the authors describe how stimulating the IgMhi/CD300a+ cell subset with PPS they produces a higher level of antibodies (IgM) than stimulating the IgM+/CD300a- subset. It would be interesting to know whether patients with lower levels of the IgMhi/CD300a+ subset have lower serum IgM levels than healthy subjects who have higher IgMhi/CD300a+ cell levels. I don't know if, before taking the PBMCs, the authors were able to dose the serum levels of IgM of the subjects studied. If it had been done I would suggest the authors to add this data.

3.       In the discussion section the characteristic of the IgMhi/CD300a+ cell subset of producing IgM antibodies in response to the stimulus with PPS (pneumococcal polysaccharide) is commented. I would like to suggest to the authors to add two or three lines of comment on the receptors that would bind PPS on IgMhi/CD300a+ B lymphocytes and on the consequent molecular mechanism that triggers the relative response observed in the present study. Two or three lines of comment are enough.

4.       In the Materials and Methods section, the authors describe the antibodies used for labeling and the related protocol. Even though the gating strategy is reported in the supplementary material I would suggest adding (in short) two lines to describe the B lymphocyte subset gating strategy also in the text, which was employed in the present study.
